Response of dung beetle diversity to remediation of soil ecosystems in the Ecuadorian Amazon

Pozo-Rivera Wilmer E. 1
Quiloango-Chimarro Carlos 1 2
http://orcid.org/0000-0003-2963-6523 Paredes Xavier 1
Landívar Mario 1
Chiriboga Carlos 1
http://orcid.org/0000-0001-9610-9788 Hidalgo Daniel 3
http://orcid.org/0000-0001-7476-6250 García Karina 3
Villacís Jaime 1 jevillacis@espe.edu.ec
1 Departamento de Ciencias de la Vida y de la Agricultura, Universidad de las Fuerzas Armadas (ESPE) , Av. General Rumiñahui s/n, Sangolquí , Ecuador
2 Department of Biosystems Engineering, Universidade de São Paulo-USP , Piracicaba, São Paulo , Brazil
3 Centro de Investigación de Tecnologías Ambientales del Proyecto Amazonía Viva, Empresa Pública PETROECUADOR , La Joya de los Sachas, Orellana , Ecuador
Shrestha Jiban
Electronic publication date: 2023 Mar 14
Publication date: 2023
Volume: 11
Electronic Location ID: e14975
Received 2022 Nov 11; Accepted 2023 Feb 7
Copyright: © 2023 Pozo-Rivera et al.
Copyright year: 2023
Copyright holder: Pozo-Rivera et al.
License: This is an open access article distributed under the terms of the Creative Commons Attribution License, which permits unrestricted use, distribution, reproduction and adaptation in any medium and for any purpose provided that it is properly attributed. For attribution, the original author(s), title, publication source (PeerJ) and either DOI or URL of the article must be cited.
License URL: https://creativecommons.org/licenses/by/4.0/

Keywords: Scarabaeinae, Tropical rain forest, Degradation, Ecological restoration

Funding: The authors received no funding for this work.

==============================
Background

Efforts to alleviate the negative effects of oil spills in the Ecuadorian Amazon include remediation activities such as cleaning, reshaping, and revegetation of polluted areas. However, studies of the diversity of biological communities in these hydrocarbon-degraded ecosystems have never been carried out. Here, we evaluated the diversity of dung beetles on remediated soil ecosystems (Agricultural Soils and Sensitive Ecosystems) and on non-contaminated soils (Natural Forests and Palm Plantations).

Methodology

The study was conducted in Sucumbíos and Orellana provinces, in the Ecuadorian Amazon at four sampling sites per ecosystem type (a total of 16 sites). At each sampling site, six pitfall traps remained active for 120 consecutive h per month for 1 year.

Results

We collected 37 species and 7,506 individuals of dung beetles. We observed significant differences in mean species abundance, richness, and diversity between non-contaminated soil ecosystems and remediated soil ecosystems, with Natural Forests presenting the highest values, and Agricultural Soils the lowest values. Regarding sampling month, we also found significant differences among ecosystems, which were also higher in Natural Forests.

Discussion

The results suggest that hydrocarbon-degraded ecosystems tend to conserve lower beetle diversity one year after remediation highlighting the importance of Natural Forests for the conservation of tropical biodiversity. Therefore, dung beetle diversity could be used for future landscape management of these hydrocarbon-degraded ecosystems.

Introduction

The extraction and transport of hydrocarbons occur worldwide on a daily basis and can lead to environmental pollution. Accidental oil spills are one of the most serious issues during these processes, contaminating soil and water sources and causing significant damage to ecosystems (Sajna et al., 2015; Souza, Vessoni-Penna & De Souza Oliveira, 2014). Chilvers, Morgan & White (2021) analyzed 1,702 oil spill events worldwide between 1970 and 2018, and reported that significant mortality of birds, reptiles, and mammals occurred in 18% of these events. Furthermore, Sharifi, Sadeghi & Akbarpour (2007) observed that pollutants affect seed germination by up to 40% in species such as Triticum sativa, Bromous mermis, Linum usitatissimum, and Agropyron deserterum.

In Ecuador, the Amazon Forest is highly affected by hydrocarbon activities (Villacís, 2016; Villacís et al., 2016a; Villacís et al., 2016b), where approximately 650,000 barrels of crude oil have been spilt between 1972 and 2015, negatively impacting the diversity of flora and fauna (Yánez & Bárcenas, 2012; Ministerio del Ambiente, 2016; Rivera-Parra et al., 2020). In response, the Ecuadorian government has remediated 1,200,098 m3 of contaminated soil (Muente et al., 2022). The remediation process includes removing vegetation and collecting waste (contaminated vegetation is composted in treatment centers and reincorporated into the site at the end of remediation activities); sucking and transporting fluids (the crude oil is collected and transported to a recuperation plant); in situ soil washing with water and biodegradable surfactants; and revegetation of intervened areas (García-Villacís et al., 2021). During the soil washing process, soil analyses are conducted until the level of pollutants reaches specific permissible levels of hydrocarbons, cadmium, nickel, and lead established in Ecuadorian environmental regulations. According to the specific permissible levels of these contaminants, the soils are classified as Agricultural Soils or Sensitive Ecosystems (higher pollutant levels in Agricultural Soils; Ministerio de Energía y Minas, 2010). Revegetation varies depending on the type of ecosystem; for example, Agricultural Soils are covered with native grasses to prevent soil erosion or prepared to cultivate oil palm, maize, cassava, and other crops, whereas Sensitive Ecosystems are revegetated with native trees to form forest fragments (Villacís et al., 2016b).

García-Villacís et al. (2021) analyzed the benefits of the remediation process in Ecuador including ecosystem variables such as acidification, terrestrial-, and freshwater-eutrophication, and freshwater ecotoxicity, and concluded that remediation decreased negative impacts by 43 percent compared to non-intervened sites. However, organisms that serve as bioindicators of ecosystem alterations were not included in this study. Changes in the abundance, diversity, and composition of these organisms can elucidate the effects of environmental disturbance and subsequent remediation (Kremen, 1992; Grand et al., 2017). Little research has been conducted using bioindicators in soils contaminated with hydrocarbons. For example, in a study conducted in France, Pérès et al. (2011), concluded that variations in the diversity of earthworms (Eisenia fetida) can be used to assess the soil quality with respect to petroleum hydrocarbons and metal pollutants.

Among the most commonly used bioindicators are dung beetles (Coleoptera, Scarabaeinae), which are distributed across a wide range of geographical locations (Lumaret & Lobo, 1996; Herzog et al., 2013), have high levels of diversity (Espinoza & Noriega, 2018), and are sensitive to microclimatic changes caused by deforestation and forest fragmentation (Campos & Hernández, 2015; Davis & Sutton, 1998; Davis et al., 2001). Abundance, richness and diversity of dung beetle communities commonly decrease under environmental changes that also affect overall ecosystem health (Otavo, Parrado-Rosselli & Noriega, 2013), and fulfill important ecological functions such as redistribution of nutrients in the soil through the burial of dung, decomposition of organic matter, soil bioturbation, secondary seed dispersal, and the pollination of plants (Rangel-Acosta & Martínez-Hernández, 2017; Fernandes et al., 2019). These ecological functions provide ecosystems services to the human population, such as the biological control of livestock pests, indirect reduction of greenhouse emissions from livestock, and soil fertilization (Lumaret, Kadiri & Martínez-Morales, 2022).

Several studies have been conducted to assess the diversity of dung beetles in degraded tropical ecosystems. For example, to evaluate the effectiveness of reforestation varying ages in the northeast Brazil (Audino, Louzada & Comita, 2014), the effect of habitat alterations as consequence of ecotourism in the North of Colombia (Noriega et al., 2020), the differences between different land uses (forest corridors, forest fragments, coffee plantations and pastures) in southeastern Brazil (Costa et al., 2017), and the effect of conservation strategies in fragmented forests in the south of Mexico (Rivera et al., 2020). Dung beetle studies in the Ecuadorian Amazon have reported between 42 and 69 species, as well as changes in their diversity caused by ecosystem disturbances such as road construction, deforestation, and cattle breeding (Carpio et al., 2009; Espinoza & Noriega, 2018; Moulatlet et al., 2021). Therefore, dung beetle diversity could be used to compare hydrocarbon-degraded ecosystems.

In this study, we hypothesized that sites with soil degradation (Sensitive Ecosystems and Agricultural Soils) would have reduced dung beetle diversity when compared to non-contaminated soils (Natural Forests and Palm Plantations). To address this hypothesis, changes in abundance, richness, and diversity of dung beetle communities were evaluated in two types of remediated ecosystems and two non-contaminated ecosystems.

Materials and Methods

Ethics statement

This study was authorized under research permit 016-2018-IC-DPAO/AVS and authorization for the mobilization of specimens 005-2019-MOV-FAU-DPAO/AVS, both issued by the Ministerio del Ambiente del Ecuador.

Study area

The study was performed in the Sucumbíos (0°5′S, 76°53′W) and Orellana (0°56′S, 75°40′W) provinces in the Ecuadorian Amazon (Fig. 1). The land use of the localities is distributed in 70% for Natural Forest, 22% for crops and pastures, and 8% for urban and industrialized areas (Ministerio de Agricultura y Ganadería, 2015). The predominant climate is Humid Tropical Rainforest according to the Holdridge climate classification (Holdridge, 1967), with a mean annual temperature of 24.5 °C and heavy rainfall throughout the year (4,132 mm). Meteorological data from an automatic weather station (0°20′S, 76°52′W) located ~22 km from the most distant sampling site were used to compute monthly values of precipitation, temperature, and relative humidity (Table 1).

Figure 1 Location of the sampling sites in the Sucumbíos and Orellana provinces.

Remediated soil ecosystems (Sensitive Ecosystems and Agricultural Soils) and non-contaminated soil ecosystems (Natural Forests and Palm Plantations).

Table 1 Mean values of climatic variables in the ecosystems evaluated.

Order	Month/Year	Rainfall (mm)	Temperature (°C)	Humidity (%)	
1	February/2018r	368.8	24.4	90.8	
2	March/2018r	429.6	24.2	92.1	
3	May/2018r	372.0	24.2	93.0	
4	June/2018r	381.6	23.5	91.4	
5	July/2018d	322.2	23.9	90.0	
6	August/2018d	293.7	24.1	89.0	
7	September/2018d	248.1	24.4	88.2	
8	October/2018d	238.4	24.9	89.1	
9	November/2018d	298.9	24.9	88.5	
10	December/2018r	350.5	24.4	89.0	
11	January/2019r	376.3	24.7	89.5	
Notes:

Meteorological data from an automatic weather station (0°20′S, 76°52′W).

r High rainfall.

d Low rainfall.

The paired-site sampling per ecosystem type were established 1 year after remediation process has been completed in ecosystems previously affected by hydrocarbon activities (Sensitive Ecosystems and Agricultural Soils). In addition, two types of non-contaminated soil ecosystems were included as controls (Natural Forests and Palm Plantations). As described earlier, Sensitive Ecosystems had lower pollutant levels than the Agricultural Soils (Table 2). Further, our Sensitive Ecosystems were beside fragmented Natural Forests and were revegetated with native trees as described by Villacís et al. (2016a) (Fig. 2A), whereas one of the Agricultural Soils was beside a Palm Plantation and three were in deforested areas (probably to be used later for cattle raising or cropland areas) (Fig. 2B). We delimited four sites (1 ha) into fragmented Natural Forests of around 400 ha (Fig. 2C) and delimited four sites (1 ha) into Palm Plantations (8 years) of the Instituto Nacional de Investigaciones Agropecuarias (Fig. 2D), both without a history of oil spills or remediation activities.

Table 2 Soil characteristics of the remediated soil ecosystems and non-contaminated soil ecosystems in the Ecuadorian Amazon.

Variable	Remediated soil ecosystems	Non-contaminated soil ecosystems	
Sensitive Ecosystems	Agricultural Soils	Natural Forests	Palm Plantations	
	Application of air and water to the soils with compressors and high-pressure pumps to release crude oil (Ministerio del Ambiente, 2023)			
Remediation activities	Yes	Yes	No	No	
Plot size per site (ha)	1	1	1	1	
Total hydrocarbons (mg kg−1)	<1,000	<2,500	0	0	
Polycyclic aromatic hydrocarbons (mg kg−1)	<1	<2	0	0	
Cadmium (mg kg−1)	<1	<2	0	0	
Nickel (mg kg−1)	<40	<50	0	0	
Lead (mg kg−1)	<80	<100	0	0	
Agrochemical use	No	No	No	Yes (herbicides and fungicides)	
Tree cover presence	Yes	No	Yes	No	
Number of present strata	2	0	4	2	
Most abundant tree species	Otoba parvifolia, Guarea sp., Pouroma sp.	No	Ceiba pentandra, Otoba parvifolia, Pouteria aubrevillei, Inga sp., Nectandra guadaripo. Cordia alliodora.	Elaeis guianensis	
Mean number of trees DBH >10 cm per ha	3.6	–	21.47	143	
DBH mean ± SE (cm)	44.13 ± 1.69	-	45.72 ± 1.61	35.45 ± 4.54	
Mean total height ± SE (m)	14.54 ± 0.65	–	16.23 ± 0.25	18.45 ± 2.21	
Note:

DBH, diameter at breast height; SE, standard error.

Figure 2 Photos from remediated soil ecosystems: (A) Sensitive Ecosystem and (B) Agricultural Soil; and non-contaminated soil ecosystems: (C) Natural Forest and (D) Palm Plantation.

Sampling design and dung beetle trapping

Sixteen sample sites (four per ecosystem type) were established within a ~30 km radius (Fig. 1). In the Natural Forests and the Palm Plantations, the sampling sites (1 ha, 100 m × 100 m) were established 200 m from the border, whereas the remediated ecosystems had irregular geometry but the sampling area was also 1 ha. Before dung beetle sampling, a soil sample from the layer 0.0–0.2 m was collected next to each pitfall trap and later mixed to form a composite soil sample per site (16 in total). To determine the total petroleum hydrocarbon and polycyclic aromatic hydrocarbon concentrations (Table 2) GC2 014 gas chromatographs were used (Shimadzu Scientific Instruments, Inc., Kyoto, Japan). In addition, the concentration of cadmium (Cd), nickel (Ni), and lead (Pb) in soils (Table 2) was determined by using atomic absorption spectrometry (AA-6800; Shimadzu Scientific Instruments, Inc., Columbia, MD, USA) as indicated by the EPA SW-846 method (Le Blanc & Majors, 2001). The analyses were performed at the Soil Laboratory of the Universidad de las Fuerzas Armadas, Ecuador.

In each sampling site, six pitfall traps were baited with pig dung and installed in the center of each sampling site separated 10 m from each other. Pitfall traps consisted of plastic containers of 0.8 L (15 cm depth × 10 cm diameter) buried up to their rims in the soil and containing solution 50:50 of water with alcohol. The traps remained active at the sites for 120 consecutive h per month during 1 year (February 2018 to January 2019). The amount of dung per trap was ~50 g and was replaced every 24–36 h. In April 2019, the traps were not evaluated due to conflict with the landowners, who prevented the entrance to the sampling sites in La Joya de los Sachas locality.

Dung beetles were preserved in 70% ethanol and identified to species-level using dichotomous keys (Chamorro et al., 2018; Vaz-De-Mello et al., 2011) and voucher specimens from the Museo de Historia Natural “Gustavo Orcés” (Escuela Politécnica Nacional, Quito, Ecuador). Some specimens were pinned and deposited in the Museum of Zoological Researches (Universidad de las Fuerzas Armadas, Sangolquí, Ecuador).

Data analysis

All analyses were performed using the software INFOSTAT (Di Rienzo et al., 2008) in interface with R (R Core Team, 2013). To assess the sampling effort, species accumulation curves were created using the Clench model, which estimates the probability of finding new species as field sampling effort increases (Clench, 1979; Soberon & Llorente, 1991). In addition, the richness observed in each type of ecosystem was evaluated using the non-parametric estimator Chao 1, which is an estimator of the number of species in a community based on the number of rare species in the sample (Chao, 1984; Colwell & Elsensohn, 2014).

Abundance, richness, and the Shannon diversity index were estimated for the four studied ecosystems using pooled data per month and sampling site as recommended by Magurran (1998). The Shannon diversity index takes into account the proportion of each species in an ecosystem studied and has been postulated as a unifying measure for the partitioning of diversity at multiple levels (Konopiński, 2020).

Abundance, richness, and diversity were also analyzed using repeated measures (by month). Differences between ecosystems were analyzed using analysis of variance with mixed models for a complete randomized design with four replications. In addition, we performed orthogonal contrasts. The first contrast evaluated differences in abundance, richness, and diversity between remediated soil ecosystems and non-contaminated soil ecosystems. The second contrast evaluated the differences between Agricultural Soils and Palm Plantations, and the third contrast evaluated the differences between Sensitive Ecosystems and Natural Forests. Furthermore, we tested differences between ecosystems, months, and interactions by using the post hoc test of Di Rienzo, Guzmán and Casanoves (DGC) (P < 0.05). The normality of the data was verified using the Shapiro-Wilks test, and the homoscedasticity was modeled using independent variances.

The Sørensen index was used to compare the similarity of dung beetle species composition between each type of ecosystem evaluated. Finally, a dendrogram was prepared using this information (Beals, 1984).

Results

Diversity of dung beetles

We collected 7,506 individual beetles of the Scarabaeinae subfamily, belonging to 13 genera and 37 species (Table 3). Abundance varied greatly, ranging from one to 1,502 individuals (an average of 202.9 individuals ± 52.8 SE per species). Canthon aequinoctialis (20% of total abundance), Ontherus sulcator (13%), Dichotomius ohausi (10%), and Deltochilum howdeni (9%), accounted for 52% of all individuals collected. Fifty-one percent of the species were classified as rare, with a relative frequency of less than ten percent. Twenty-two percent of the total species collected were found in all four evaluated ecosystems, while 12 others were exclusive to one of them: 11 in the Natural Forest and one in the Sensitive Ecosystem. Canthidium aurifex, Eurysternus atrosericus, Ontherus sulcatur, Onthophagus osculatii, and O. nyctopus are new provincial records, whereas O. hircus is a new record for Ecuador (Table 3).

Table 3 Dung beetle species collected in remediated soil ecosystems and non-contaminated soil ecosystems in the Ecuadorian Amazon.

No.	Species	Record type	Agricultural Soils	Natural Forests	Sensitive Ecosystems	Palm Plantations	Assemblage	
1	Canthon aequinoctialis Harold, 1868	RE		1,490	12		1,502	
2	Ontherus sulcatur Fabricius, 1775	NR-P	18	135	747	90	990	
3	Dichotomius ohausi Luederwaldt, 1923	RE	7	639	17	72	735	
4	Deltochilum howdeni Martínez, 1955	NR-P		671	13	1	685	
5	Onthophagus haematopus Harold, 1887	RE		493	7		500	
6	Eurysternus plebejus Harold, 1880	RE	6	386	4	30	426	
7	Onthophagus osculatii Guérin-Méneville, 1855	NR-P		395	7		402	
8	Coprophaneus telamon Erichson, 1847	RE	11	307	22	33	373	
9	Onthophagus xanthomerus Bates, 1887	RE		246	22	5	273	
10	Dichotomius sp. 1	NA		232		1	233	
11	Dichotomius mamillatus Felsche,1901	RE		186	2	1	189	
12	Deltochilum amazonicum Kolbe, 1905	RE		175		3	178	
13	Eurysternus atrosericus Génier, 2009	NR-P	1	134	5	5	145	
14	Eurysternus squamosus Génier, 2009	RE		107			107	
15	Canthon luteicoliis Erichson, 1847	RE		92	1	1	94	
16	Eurysternus caribaeus Herbst, 1789	RE	4	76		13	93	
17	Eurysternus wittmerorum Martínez, 1988	RE	1	61	7	10	79	
18	Canthidium cf. rufinum Harold, 1867	RE		76			76	
19	Onthophagus hircus Billberg, 1815	NR-E	10		40	6	56	
20	Onthophagus nyctopus Bates, 1887	NR-P		52	2		54	
21	Dichotomius podalirius Felsche, 1901	RE		49			49	
22	Uroxys sp. 1	NA		48			48	
23	Eurysternus foedus Guérin-Méneville, 1844	RE	3	30	1	6	40	
24	Phaneus chalcomelas Perty, 1830	RE		28	3	1	32	
25	Scyballocanthon macullatus Schmidt, 1920	RE		31	1		32	
26	Oxysternon silenus d’Olsouefieff, 1924	RE	1	4	1	16	22	
27	Eurysternus hamaticollis Balthasar, 1939	RE		19	1	1	21	
28	Onthophagus onore Zunino & Halffter, 1997	RE		15			15	
29	Canthidium sp. 1	NA		10	3	1	14	
30	Canthidium aurifex Bates, 1887	NR-P		9	3		12	
31	Oxysternon conspicillatum Weber, 1801	RE		8			8	
32	Deltochilum carinatum Westwood, 1837	RE		7			7	
33	Onthophagus marginicollis Harold, 1880	RE			6		6	
34	Scyballocanthon furvus Schmidt, 1920	NA		4			4	
35	Canthidium onitoides Perty, 1830	RE		3			3	
36	Malagoniella astyanax Halffter, Pereira & Martínez, 1960	RE		2			2	
37	Canthon angustatus Harold, 1867	RE		1			1	
Abundance	62	6,221	927	296	7,506	
Richness	10	35	23	19	37	
Note:

Data in bold indicate species with greatest specific abundance by ecosystem type in the study. RE, registered in the provinces studied, NR-P, newly registered in the provinces studied, NA, not evaluated, NRE, newly registered in Ecuador.

Difference in diversity between habitats over sampling time

As the sampling time increased, the species accumulation curve stabilized after the 7th month in the Natural Forests, whereas in the Sensitive Ecosystems, Agricultural Soils and Palm Plantations, the slope did not reach values close to 0 (Fig. 3). The non-parametric estimator of richness (Chao 1) showed that 99% of the species in Natural Forest, 97.63% in Sensitive Ecosystems, 94.34% in Agricultural Soils, and 87.35% in Palm Plantations were recorded. Cluster analysis showed that Agricultural Soils and Palm Plantations presented a species similarity of 32.4%, and both were similar to the Sensitive Ecosystems at 26.8%. The Natural Forests were similar to other ecosystems only at 8.53% (Fig. 4).

Figure 3 Species accumulation curves of Scarabaeinae communities recorded in remediated soil ecosystems and non-contaminated soil ecosystems in Ecuadorian Amazon.

Remediated soil ecosystems (Sensitive Ecosystem and Agricultural Soil) and non-contaminated soil ecosystems (Natural Forest and Palm Plantation).

Figure 4 Species similarity clusters based on the Bray-Curtis distance of the Sorensen similarity percentage in remediated soil ecosystems and non-contaminated soil ecosystems in the Ecuadorian Amazon.

Remediated soil ecosystems (Sensitive Ecosystem and Agricultural Soil) and non-contaminated soil ecosystems (Natural Forest and Palm Plantation).

The average values of abundance, richness, and the Shannon index differed between ecosystems within each month (there was a significant statistical interaction ecosystem × months; P = 0.0073; Table 4). The orthogonal contrast showed that non-contaminated soil ecosystems contained higher abundance, richness and diversity of beetles in comparison to remediated soil ecosystems (F1, 132 = 313.51, P < 0.0001). Natural Forest presented higher abundance, richness and diversity than Sensitive Ecosystems (F1, 132 = 313.51, P < 0.0001) and Palm Plantations presented higher abundance, richness and diversity than Agricultural Soils (F1, 132 = 51.60, P < 0.0001; Fig. 5).

Table 4 Analysis of variance for the abundance, richness, and diversity of beetles that were collected monthly in remediated soil ecosystems and non-contaminated soil ecosystems in the Ecuadorian Amazon.

Source	df	Abundance	Richness	Shannon	
F	P	F	P	F	P	
Ecosystems types	(3)	302.09	<0.0001	462.21	<0.0001	314.65	<0.0001	
Remediated soil ecosystems vs Control ecosystems	1	374.32	<0.0001	483.13	<0.0001	313.51	<0.0001	
Agricultural Soils vs Palm Plantations	1	991.54	<0.0001	59.97	<0.0001	51.60	<0.0001	
Sensitive Ecosystems vs Natural Forests	1	289.20	<0.0001	502.33	<0.0001	427.16	<0.0001	
Months	10	26.09	<0.0001	18.63	<0.0001	9.56	<0.0001	
Ecosystems × Months	30	8.07	<0.0001	2.41	0.0003	1.90	0.0073	
Notes:

Ecosystems were considered fixed factor and months random factor. Remediated soil ecosystems (Sensitive Ecosystems and Agricultural Soils) and non-contaminated soil ecosystems (Natural Forests and Palm Plantations).

df, degrees of freedom; F, result of F-Fisher value; P, result of probability.

Figure 5 Abundance, richness, and diversity of beetles that were collected in remediated soil ecosystems and non-contaminated soil ecosystems in the Ecuadorian Amazon.

Bars are means ± standard error (n = 44; four data from each ecosystem type × 11 months of sampling). Remediated soil ecosystems (Sensitive Ecosystem and Agricultural Soil) and non-contaminated soil ecosystems (Natural Forest and Palm Plantation).

The highest monthly average values for abundance (January and November), richness (January, February, September, October and November) and Shannon’s index (September and November) were recorded in the Natural Forests (Fig. 6). Meanwhile the average values of abundance, richness and diversity for Sensitive Ecosystems, Palm Plantations and Agricultural Soils were lower than in the Natural Forests, showing little variation during the 11 months of sampling.

Figure 6 Abundance (top), richness (middle), and diversity (bottom) of beetles collected monthly in remediated soil ecosystems and non-contaminated soil ecosystems in the Ecuadorian Amazon.

Values are means ± standard error (n = 4; four data from each ecosystem type). Different letters in each point indicate significant differences (DGC, P < 0.05). Remediated soil ecosystems (Sensitive Ecosystem and Agricultural Soil) and non-contaminated soil ecosystems (Natural Forest and Palm Plantation).

Discussion

Dung beetle diversity

The species presented in this study represent 17% of the 220 species of dung beetles registered in Ecuador (Chamorro et al., 2018) and more than 50% of previously registered species in the Orellana and Sucumbíos provinces. Five species are new for these provinces, whereas Ontophagus hircus was recorded for the first time in Ecuador. This demonstrates that beetle diversity must be studied not only to understand the effects of ecosystem disturbance but also to complete the species inventory of the Ecuadorian Amazon.

The stabilization of the species accumulation curves in the Natural Forest supports the registration of almost all species in that ecosystem. As for the other ecosystems (Sensitive Ecosystems, Agricultural Soils, and Palm Plantations), the slope of the curves suggests that we did not sample the total species richness of dung beetle communities (Gering, Crist & Veech, 2003).

The community structure in the non-contaminated soil ecosystems tended toward high abundance, richness, and diversity when compared to the remediated soil ecosystems, indicating that the ecosystems has likely not fully recovered. According to a recent study, dung beetles bioaccumulate more Pb than other insects such as cicada and longicorn (Zhou et al., 2019). In another Coleoptera family, the Carabidae, modifications of the digestive tract were reported in individuals exposed to concentrations of 10 mg kg−1 Pb (Conti et al., 2017). In addition, the presence of heavy metals in soils negatively influences small mammals (Richardson et al., 2015), which decreases food sources for dung beetles. These factors may negatively affect dung beetle diversity in hydrocarbon-degraded ecosystems, but further research should be carried out to elucidate the effects of other oil pollutants found in the studied ecosystems on these organisms.

Similarly, cluster analyses showed an acute division between the Natural Forests and other ecosystems evaluated. This could be because Natural Forest fragments within human modified landscapes constitute wildlife refuges (Blaum et al., 2009). This trend of decreased dung beetle diversity between the Natural Forest and the other ecosystem types follows a general decreasing gradient of diversity and an increase in anthropomorphic disturbances due to contamination and land use (Aninta et al., 2019; McCain, 2005; Stevens & Gavilanez, 2015). For example, Scarabaeinae diversity in Palm Plantations was similar to that found in Agricultural Soils. This is consistent with previous studies of Fitzherbert et al. (2008) and Harada et al. (2020), which reported that agrochemicals could favor the degradation of soil and nutrients and hence diminish dung beetle diversity.

Habitat and temporal variation of dung beetle diversity

Dung beetles are very sensitive to habitat disturbance (Audino, Louzada & Comita, 2014; Campos & Hernández, 2015; Da Silva, Vaz-de-Mello & Di Mare, 2013). For example, changes in dung beetle community structure have been reported under low vegetation cover (Nichols et al., 2007) as a result of intense solar radiation on the soil surface, which accelerates the decomposition rate of food sources (Méndez et al., 2019). Moreover, chemical perturbations affect dung beetle communities in several ways such as changes in composition, diversity and population (Hutton & Giller, 2003; Correa et al., 2022). Therefore, the presence of hydrocarbons and heavy metals in Agricultural Soils and Sensitive Ecosystems as well as landscape modifications at oil exploitation sites could influence abundance, richness, and diversity of dung beetle assemblages.

The diversity of dung beetles is determined by regional rainfall patterns (Novais et al., 2016). Our results indicated that the diversity of dung beetles in the Natural Forests was higher during the month with lower levels of precipitation (October 238 mm month−1) which is consistent with the study of Ibarra-Polesel, Damborsky & Porcel (2015), which investigated dung beetles in subtropical ecosystems. However, in contrast, several other studies have demonstrated that higher beetle diversity is linked to the months with elevated values of precipitation (Escobar et al., 2008; Nunes et al., 2016; Rangel-Acosta & Martínez-Hernández, 2017).

We may have found higher dung beetle diversity during the months with the lowest levels of rainfall due to the interference of other environmental factors. For example, alteration of microclimates and microhabitats (Medina, Escobar & Kattan, 2002; Noriega & Realpe, 2018; Sánchez-Hernández et al., 2022), changes in trophic structure (Novais et al., 2016), as well as altitudinal gradient effects (Noriega & Realpe, 2018) could have affected the mobility, displacement, and genetic flow of organisms between ecosystems (Harvey, Gonzalez & Somarriba, 2006). The Natural Forests in the provinces where the study was conducted are under a higher degree of environmental disturbance (deforestation, forest fragmentation, oil spills, population growth, etc.) (Rivera-Parra et al., 2020) and, in general, the entire Amazon basin is under massive degradation due to deforestation (Marin et al., 2022).

Implications for the conservation

Our study provides the first quantitative data on dung beetle communities in ecosystems affected by hydrocarbon activities in the Ecuadorian Amazon. The diversity of dung beetles provides a useful tool for assessing the temporary status of remediated sites previously affected by hydrocarbon activities. Although differences in beetle diversity were found between remediated ecosystems, similar recommendations for conservation measures can be made for both Agricultural Soils and Sensitive Ecosystems. Therefore, Agricultural Soils and Sensitive Ecosystems should not only be based on the levels of hydrocarbons and heavy metals but also on the diversity of bioindicators.

The presence of dung beetles in remediated ecosystems provides a baseline for implementing strategies to improve the existing diversity. However, the conservation of biodiversity in remediated ecosystems depends not only on remediation activities, but also on other anthropogenic activities in the Amazonian tropical forests (Rivera-Parra et al., 2020).

Conclusions

This research show that in hydrocarbon-degraded ecosystems, dung beetle abundance, richness, and diversity were lower compared to sites that had not been affected by oil spills. Dung beetle diversity changed throughout the year and was significantly higher in months with little rainfall. Our findings provides a baseline that could be used for conservation planning and management of hydrocarbon- and heavy metal-contaminated ecosystems.

Supplemental Information

Supplemental Information 1 Data.

The general data for all the dung beetles collected: the species collected, the type of ecosystem, the repetition, the specific abundance, and the month of collection. We used this sheet to obtain table 3 and to extract the monthly abundance, richness, and diversity index. In the diversity index sheet, we present the monthly variation of the abundance, richness, and diversity of Shannon: the ecosystems, the repetition, the moths, the richness, the abundance, and the Shannon index. With this data, we performed the analysis of variance and the comparison of means tests, and we elaborated table 4, and figures 2, 3, 4, and 5.

Click here for additional data file.

The authors would like to thank entomologists Wladimir Carvajal (Escuela Politécnica Nacional), Geovanny Onore (Otonga Foundation), and Juan Tigrero-Salas (Universidad de las Fuerzas Armadas) for their help with identifying dung beetle species. To Juan Carlos López for his valuable help in collecting field data and logistical support. The authors would also like to thank Petroamazonas EP and Universidad de las Fuerzas Armadas for the logistical and financial support provided to carry out this research.

Additional Information and Declarations

Competing Interests

Author Contributions

Field Study Permissions

Data Availability

Karina García is employed by Empresa Pública PETROECUADOR.

Daniel Hidalgo was an employee of Empresa Pública PETROECUADOR during the development of the research.

Wilmer E Pozo-Rivera conceived and designed the experiments, performed the experiments, analyzed the data, prepared figures and/or tables, authored or reviewed drafts of the article, and approved the final draft.

Carlos Quiloango-Chimarro conceived and designed the experiments, performed the experiments, authored or reviewed drafts of the article, logistical Support, and approved the final draft.

Xavier Paredes conceived and designed the experiments, performed the experiments, authored or reviewed drafts of the article, logistical Support, and approved the final draft.

Mario Landívar conceived and designed the experiments, performed the experiments, authored or reviewed drafts of the article, logistical Support, and approved the final draft.

Carlos Chiriboga conceived and designed the experiments, performed the experiments, authored or reviewed drafts of the article, logistical Support, and approved the final draft.

Daniel Hidalgo conceived and designed the experiments, prepared figures and/or tables, logistical Support, and approved the final draft.

Karina García conceived and designed the experiments, prepared figures and/or tables, logistical Support, and approved the final draft.

Jaime Villacís conceived and designed the experiments, performed the experiments, analyzed the data, prepared figures and/or tables, authored or reviewed drafts of the article, logistical Support, and approved the final draft.

The following information was supplied relating to field study approvals (i.e., approving body and any reference numbers):

Field experiments were approved by the Ministerio del Ambiente of Ecuador for issuing research permit No. 016-2018-IC-DPAO/AVS and authorization for the mobilization of specimens No. 005-2019-MOV-FAU-DPAO/AVS.

The following information was supplied regarding data availability:

The raw data is available as a Supplemental File.

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
