# Peer review of "Response of dung beetle diversity to remediation of soil ecosystems in the Ecuadorian Amazon"

_PeerJ, doi:10.7717/peerj.14975_

## Round 0.1 · original submission · Major Revisions

Dear Authors

Kindly include reviewers' comments step by step in your manuscript before re-submission. And also make the English editing before re-submission. Thank you.

Editor

Reviewer 1 ·

Basic reporting

Abstract
Line 23 give an example of hydrocarbon activities for clarity eg oil extraction, also mention that this causes environmental disturbance.
Line 26 give example of how the soil is remediated for clarity.
Line 29 “monitored monthly for a 120-hour period during one year.” Unclear whether this was a 120 hour period per month or throughout the year. Make clearer.
Line 34 Is this remarkable given that they are similar types of ecosystem? Eg highly managed. Or is this not true of the agricultural habitat type. Throughout what these terms mean needs to be made clearer.
Line 36 Unsure if you can claim this final line. How could the remediation process be improved using these data?
Introduction
Line 47 Too broad to claim “all life forms”. Give specific examples. It would be nice to see some studies which give examples of species that are harmed by hydrocarbons.
Line 53 reference should not be in caps.
Line 53 Start new paragraph to discuss the remediation process. Go into further detail what this involves.
Line 55 Define agricultural soil and sensitive ecosystem. It's not clear which is more contaminated or if they were different sorts of ecosystem to begin with. As above, this needs to be made clearer throughout.
Line 59 I would like to see more detail for the García-Villacís et al. (2021) study.
Generally I feel like more detail is needed in the intro. What other species have been used as bioindicators of hydrocarbons? A paragraph about this would be nice.
Line 73 Again, more detail from these studies would be beneficial.
Line 76 “It was hypothesized that sites with soil degradation (Sensitive ecosystems and Agricultural soils) decrease dung beetle diversity when compared to non-contaminated soils (Natural forests and Palm plantations).” “decrease” should be “would have reduced”.
Methods
Line 96 Capital for Table 1.
Line 100 Again, I would like to see a definition for the different ecosystem types eg the total hydrocarbons from Table 2. Was there any difference between these sites before contamination took place? Make very clear which type has the lower hydrocarbon levels. Also how long ago did remediation take place for these sites? Would we expect dung beetle diversity to be increasing over the year due to beetles being able to live in these sites again?
Results
Structure of results unclear at times eg line 170 – a paragraph should not be one sentence long. Generally the results could do with restructuring so that they’re easier to understand.
165-167 Not clear what this means. Define accumulated richness earlier in the paper. Surely if accumulated it would always go up with time?
168 Also not sure what this means – what are inventory efficiencies? Define in methods.
172 – this is the main result and should be higher up in the results section. Keeping one idea to a paragraph would make it easier for the reader to understand eg could use the subheadings that you used in the discussion.

Figs/Tables
Figure 1 – colouring the legend by contaminated or non contaminated would help with understanding (as coloured in blue and red below).
Figure 2 – Would have been nice if these symbols matched those on the map.
Figure 3 – colour of key is different to colour of bar plot.
Table 2 – I would like more of this to be brought into the main text.
Raw data have been shared - thanks!

Experimental design

Line 75 I don't see how this study investigates the impact of remediation. If you were going back to the same sites before and after remediation I would agree, but you are just comparing 4 different types of site eg they differ in tree cover, location etc. Sites aren’t paired on similarity. So I feel the aims of the paper need changing. I feel like you can say it’s a comparison between sites which have been degraded by hydrocarbon activities and those which haven’t but you need to state more clearly in what ways they are the same so that the reader can be more confident that this is due to hydrocarbon activities and not just existing differences between the sites.
Line 103 – Need more explanation of why these are suitable as controls. Eg how were they similar to the contaminated sites.
Line 115 How did you chose the sites for the six pitfall traps? Add to methods.
Line 118 Not clear whether the 120 hours is all at once or split over the month.
Line 131 define inventories
Line 136 define Shannon diversity index.
Line 137 “Pooled data were used to compare the four types of ecosystems due to the homogeneity among the four sampling sites per ecosystem type.” Not sure what you mean by this, please expand.
Line 147 DCG - please write in full.

Validity of the findings

202 Da Silva found high dung beetle diversity in natural forests which was also your finding. So I concerned that its the habitat type, rather than the hydrocarbons which are causing the difference.
213 it should be made clearer earlier in the paper that landscape modifications also came into the reason for these results.
262 I don’t think you can make this claim. Yes diversity was different between the sites but other factors as well as hydrocarbon presence were different too. As stated above in the “Experimental design” section.
I would be much more convinced if the amount of hydrocarbon found in the soil at each site was added to your models, finding decreased dung beetle diversity with increasing hydrocarbons.
Data looks good.

Additional comments

Overall, I enjoyed reading this manuscript and would like to see it published given the corrections I have outlined.
Generally, I would like to see more detail in the introduction and discussion, expand on your points and go into further detail when describing studies. Quite a few terms also need defining to aid the readers understanding.
I think the main finding of the study needs to be toned down or modified. Yes, I am convinced that there was a difference in diversity between the habitat types. I’m not convinced that the data shows that this is necessarily due to differences in hydrocarbon levels. It could be due to tree coverage for example changing the types of mammals present and therefore food for the dung beetles. To remedy this a better definition of the habitat types is needed and why the controls are suitable. If the quantity of hydrocarbons found in the soil could be added to the models this would be convincing.

·

Basic reporting

The manuscript entitled “Dung Beetles Diversity to Remediated Soils Ecosystems in the Ecuadorian Amazon” compares the diversity, richness and abundance of dung beetles in four types of ecosystems: Agricultural soils and Sensitive ecosystems (as remediated soils ecosystems) and Natural forest and Palm plantations (as non contaminated soils ecosystem). The number of species and the number of individuals collected is relevant, and the authors find large differences in these values between the natural forest and the rest of the ecosystems. In addition, with their data they provide five new provincial records, and a new species record for Ecuador.

All the literature references are relevant, and correctly listed.

The structure of the article conforms to the suggested format of PeerJ.

The data on which the conclusions are based are provided in the Supplementary Material section.

I think English is well written and is understandable.

Experimental design

The sampling has been carried out for practically almost one year, having collected a very high and representative number of individuals and species of the study area. I think that the sampling sites are well chosen and the sampling methodology is adequate.

In the statistical part I miss a little a better justification and explanation of the analyzes carried out.

Validity of the findings

The data from this work leave no doubt about the great impact that anthropic practices have on ecosystems for dung beetles.

Taking into account that dung beetles are considered bioindicator species due to their sensitivity to changes in the environment, this paper could be useful for decision-making about biodiversity conservation programs.

In addition, it updates the inventory of species in the sampled areas.

Additional comments

The article is interesting and I think the data obtained is relevant, although in general I think there is a little lack of detail describing some things about the material and methods or results. I also encourage the authors to discuss their findings more specifically.

I have written some comments with the intention of contributing ideas that can help the authors to include some aspects. Of course, this is my opinion and all of it is debatable.

I attached a PDF.

---

## Round 0.2 · Minor Revisions

The authors need to include minor suggestions of both reviewers.

Reviewer 1 ·

Basic reporting

Overall, I think the changes you have made have greatly improved the MS. I have made some further changes in the attached track changes document to make some of the language clearer. I am happy for this paper to be published after these changes have been made.

Experimental design

Abstract –The abstract reads well.
Line 37 – how long after remediation? Have added “one year after remediation” to make clear. Eg dung beetle diversity could improve in the future.
Line 46 I still don’t like ”in all life forms”. Instead “causing significant damage to ecosystems” would be better. I have made this change in the tracked changes doc.
Line 53 most affected site by hydrocarbon activities – most affected compared to what? Have changed to highly affected. 650,000 barrels of crude oil have been spilt – over what time period?
Line 64 – I have changed to “Soils are classified as agricultural soil or sensitive ecosystems” write what the classification depends on eg according to flora and fauna at the site, or what the site was used for before the oil spill.
Line 86 “commonly decrease” decrease in what – abundance, species richness – be specific.
Line 102 – Add a little more to introduce the study in the final paragraph eg where the study was conducted.
Section “Selection of collection sites” Still could be clearer. Eg do you mean that some of the sites were paired?
Capitalisations of sampling site types throughout eg Sensitive Ecosystems
Line 235 “For all evaluated ecosystems, the richness of dung beetle communities was greater than 87%, which suggests that minimum changes in the richness could exist if sampling effort is increased” Not sure what this means, please reword.
Line 242 “This demonstrates that beetle diversity must be studied on disturbed soils in tropical forests” Why? Were these species found in the disturbed soils?

Validity of the findings

NA

Additional comments

NA

Annotated reviews are not available for download in order to protect the identity of reviewers who chose to remain anonymous.

·

Basic reporting

I am grateful to the authors for considering and applying many of the suggestions I have made.

I only make a few small comments without much importance. They are in red in the attached document.

Experimental design

No comment

Validity of the findings

No comment

Additional comments

No comment

---

## Round 0.3 · accepted · Accept

The manuscript is acceptable.

The ASection Editor noted:

> I would suggest changing the "dung beetles" in the title to the singular "dung beetle" as the term is acting as a modifier to "diversity". Otherwise it should be "dung beetles'" (i.e. possessive), but I don't think that's the authors' intent.